# Immunization of turkeys with *Clostridium septicum* alpha toxin-based recombinant subunit proteins can confer protection against experimental Clostridial dermatitis

Feba Ann John[1], Valeria Criollo[1], Carissa Gaghan[1], Abigail Armwood[1], Jennifer Holmes[2], Anil J. Thachil[3], Rocio Crespo[1], Raveendra R. Kulkarni[1]*

1 Department of Population Health and Pathobiology, College of Veterinary Medicine, North Carolina State University, Raleigh, North Carolina, United States of America, 2 Department of Clinical Sciences, College of Veterinary Medicine, North Carolina State University, Raleigh, North Carolina, United States of America, 3 Bacteriology & Mycology Division, Rollins Animal Disease Diagnostic Laboratory, Raleigh, North Carolina, United States of America

* rrkulkar@ncsu.edu

**Data Availability Statement:** All relevant data are within the manuscript and its Supporting Information files.

## Abstract

Clostridial dermatitis (CD), caused by *Clostridium septicum*, is an emerging disease of increasing economic importance in turkeys. Currently, there are no effective vaccines for CD control. Here, two non-toxic domains of *C. septicum* alpha toxin, namely ntATX-D1 and ntATX-D2, were identified, cloned, and expressed in *Escherichia coli* as recombinant sub-unit proteins to investigate their use as potential vaccine candidates. Experimental groups consisted of a Negative control (NCx) that did not receive *C. septicum* challenge, while the adjuvant-only Positive control (PCx), ntATX-D1 immunization (D1) and ntATX-D2 immunization (D2) groups received *C. septicum* challenge. Turkeys were immunized subcutaneously with 100 μg of protein at 7, 8 and 9 weeks of age along with an oil-in-water nano-emulsion adjuvant, followed by *C. septicum* challenge at 11 weeks of age. Results showed that while 46.2% of birds in the PCx group died post-challenge, the rate of mortality in D1- or D2-immunization groups was 13.3%. The gross and histopathological lesions in the skin, muscle and spleen showed that the disease severity was highest in PCx group, while the D2-immunized birds had significantly lower lesion scores when compared to PCx. Gene expression analysis revealed that PCx birds had significantly higher expression of pro-inflammatory cytokine genes in the skin, muscle and spleen than the NCx group, while the D2 group had significantly lower expression of these genes compared to PCx. Peripheral blood cellular analysis showed increased frequencies of activated CD4+ and/or CD8+ cells in the D1 and D2-immunized groups. Additionally, the immunized turkeys developed antigen-specific serum IgY antibodies. Collectively, these findings indicate that ntATX proteins, specifically the ntATX-D2 can be a promising vaccine candidate for protecting turkeys against CD and that the protection mechanisms may include downregulation of *C. septicum*-induced inflammation and increased CD4+ and CD8+ cellular activation.

**Funding:** This work was funded to R.R.K. by the US Poultry and Egg Association foundation grant (# BRF-014) and the North Carolina State University-Animal Health and Nutrition Consortium (grant# unavailable). https://www.uspoultry.org/ https://sites.google.com/ncsu.edu/afnc/home Funders have not played any role in the study design, data collection and analysis, decision to publish, or preparation of the manuscript.

**Competing interests:** US Provisional Patent Nos. 63/609,022, 63/608,491, 63/597,112 titled "Vaccine Composition Against Clostridial Dermatitis". This does not alter our adherence to PLOS ONE policies on sharing data and materials. The sequences of ntATX-D1 and ntATX-D2 vaccine antigens reported in this work are also deposited in the GenBank (https://www.ncbi.nlm.nih.gov/ nuccore/PP003322 and https://www.ncbi.nlm.nih. gov/nuccore/PP003323).

## Introduction

Clostridial Dermatitis (CD), also referred to as Gangrenous dermatitis or Cellulitis, is an emerging economically devastating disease of poultry with its prevalence on the rise in the past decade or so [1,2]. In turkeys, CD usually peaks around 13–18 weeks of age and is characterized by subcutaneous edema and emphysema with skin lesions in the breast/inguinal area, sometime extending to underlying muscle tissue followed by necrotic dermatitis and sudden death [3]. Although several etiological agents have been implicated, the two known Clostridial pathogens are *C. septicum* and *C. perfringens* [2–4]; however, *C. septicum* has more often been isolated and identified as the primary etiological agent for CD in turkeys [1,5–7]. Predisposing factors such as overcrowding, poor ventilation and immunosuppression are implicated in CD development [2,4,7]. Despite good management practices coupled with antibiotic treatment, CD control has remained challenging. Unfortunately, no effective commercial vaccines are currently available for poultry, which may be due to the complexity of disease pathogenesis and a poor understanding of immunity to CD.

*Clostridium septicum* is a Gram-positive, anaerobic, spore forming and toxin-producing bacterium with alpha-toxin (ATX) implicated as the key virulence factor in the disease pathogenesis [8,9]. Two theories are currently available to explain CD pathogenesis. The 'Inside out' theory proposes that Clostridia, being ubiquitous and opportunistic in nature, can cause intestinal inflammation resulting in gut leakage, which in turn will facilitate them to gain access to circulation, and to the sites of skin abrasions or hypoxia, where they can multiply and produce toxins [4]. The 'outside in' theory suggests that the agent enters the subcutaneous tissues via skin wounds, where they multiply and produce CD. In support of the latter theory, others and we have previously shown that turkeys inoculated with *C. septicum* subcutaneously can cause clinical CD, characterized by typical dermatitis lesions and related mortality [3,6,10].

Amongst the non-antibiotic-based disease control measures, vaccination seems to offer a promising CD control strategy. Much of the vaccine work has focused on inducing antibodies against *C. septicum* ATX. Our previous work showed that turkeys immunized subcutaneously with *C. septicum* bacterin-toxoid vaccine, under both laboratory and commercial setting, had significantly higher anti-*C. septicum* antibodies and reduced morbidity and mortality [11]. In one of the two field trials, *C. septicum* toxoid vaccine reduced CD-related mortality by about 50%. More recently, another study showed that adjuvantation of *C. septicum* bacterin-toxoid with a water-in-oil Montanide emulsion administered to turkeys subcutaneously in commercial farm setting could reduce CD morbidity and mortality along with increased serum antibodies [12]. Considering the critical role of ATX in CD pathogenesis and immunity, a previous study evaluated the efficacy of a noncytolytic *C. septicum* alpha-toxin (NCAT) peptide as a potential vaccine candidate in turkeys against CD when immunized subcutaneously [8]. The immunization groups consisted of (1) Purified NCAT, (2) Crude NCAT, (3) Purified NCAT+*C. septicum* bacterin + alpha-toxoid, (4) Bacterin + alpha-toxoid, and (5) Unimmunized-challenged control. Results showed that while the group #4 was the most efficacious, others also significantly reduced mortality suggesting NCAT may be a good vaccine candidate.

In the present study, we took an approach to identify two non-toxic domains of ATX, and clone, express and purify them from *E. coli* as recombinant proteins to immunize turkeys subcutaneously and assess protection against CD using an experimental infection/challenge model. The protection assessment parameters included (1) Clinical signs, mortality, and gross pathology, (2) Histopathology of skin, skeletal muscle and spleen tissues, (3) Immune gene expression in skin, muscle, spleen and cecal tonsil (CT), (4) Peripheral blood mononuclear cell (PBMC) responses, and (5) Serum antibody evaluation.

## Materials and methods

### Cloning, expression, and purification of recombinant proteins

Based on the ATX sequence information [8,13–16], two regions were identified, which were predicted to lack cellular toxicity if cloned and expressed as recombinant proteins. The first region, referred to as non-toxic ATX Domain 1 (ntATX-D1), a 912 bp segment, was devoid of the proteolytic cleavage site (Arg-367-Ser-368), which is critical for toxin's activity (Fig 1A). The second region, referred to as non-toxic ATX Domain 2 (ntATX-D2), a 531 bp segment,

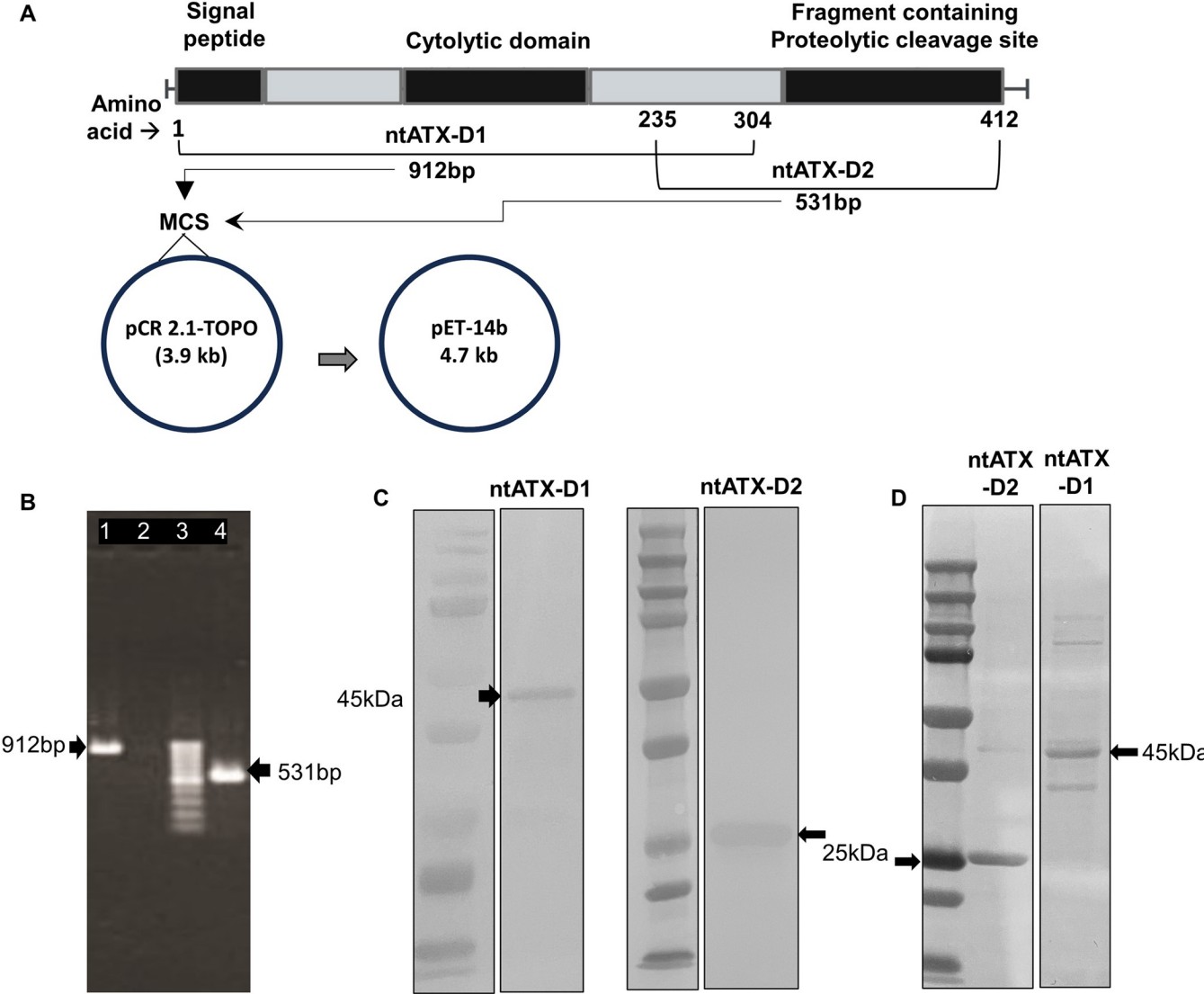

**Fig 1. Cloning of non-toxic domains of *C. septicum* alpha toxin.** Based on the alpha toxin (ATX) sequence information, two non-toxic (nt) regions devoid of critical sites for toxicity were identified; the ntATX-D1 (912 bp) devoid of proteolytic cleavage site (Arg-367-Ser-368) and ntATX-D2 (531 bp) devoid of a signal peptide sequence (residue 1–31) and the region corresponding to cytolytic, pore-forming domain of ATX (residue 203–232), as shown in panel A. Genes encoding ntATX-D1 / ntATX-D2 were amplified from *C. septicum* (panel B: Lane 1- ntATX-D1, Lane 2- Negative control, Lane 3- 100bp ladder, Lane 4-ntATX-D2). The ntATX-D1 and ntATX-D2 gene fragments were cloned first into the 'cloning' vector (pCR2.1-TA) and the *E. coli* DH5α (shuttle host) followed by cloning into pET14b expression vector between the XhoI and BamHI sites. The histidine-tagged recombinant ntATX-D1 (~45 kDa) or ntATX-D2 (~25 kDa) proteins were expressed and purified from *E. coli* BL21-DE3 (expression host) and immunoreacted in Western blot assays with anti-His antibodies (panel C) and serum from turkeys infected with *C. septicum* (panel D).

**Table 1. Primers sequences used in real-time PCR assay.**

| Target Gene | Primer Sequence (5'-3') | Annealing Temp (˚C) | GenBank Accession # |
|---|---|---|---|
| ntATX-D1 | F- AAATTCAGTGTGCGGCAGTAGTA R- CCGTTCCTCAGCACTCATACCGT | 55 | This work (PP003322) |
| ntATX-D2 | F-AAATACTGCAACAGTTTCTCCACA R-CCGTCTAAATCCTGGAACATCTTGTGT | 55 | This work (PP003323) |
| IL-1β | F-GTGAGGCTCAACATTGCGCTGTA R-GTCCAGGCGGTAGAAGATGAAG | 64 | AJ009800 |
| IL-6 | F-CGTGTGCGAGAACAGCATGGAGA R-TCAGGCATTTCTCCTCGTCGAAGC | 60 | NM_204628.1 |
| IL-10 | F-AGCAGATCAAGGAGACGTTC R-ATCAGCAGGTACTCCTCGAT | 55 | AJ621614 |
| IFNγ | F-ACACTGACAAGTCAAAGCCGCACA R-AGTCGTTCATCGGGAGCTTGGC | 60 | X99774 |
| TGFβ | F-CGGCCGACGATGAGTGGCTC R-CGGGGCCCATCTCACAGGGA | 60 | M31160.1 |
| IL-4 | F-TGTGCCCACGCTGTGCTTACA R-CTTGTGGCAGTGCTGGCTCTCC | 64 | GU119892 |
| IL-13 | F-ACTTGTCCAAGCTGAAGCTGTC R- TCTTGCAGTCGGTCATGTTGTC | 60 | AJ621250 |
| TLR 21 | F-CCTGCGCAAGTGTCCGCTCA R-GCCCCAGGTCCAGGAAGCAG | 60 | NM_001030558.1 |
| β-actin | F-CAACACAGTGCTGTCTGGTGGTA R-ATCGTACTCCTGCTTGCTGATCC | 58 | X00182 |

was devoid of a signal peptide sequence (residue 1–31) and the region corresponding to cytolytic, pore-forming region of ATX (residue 203–232) (Fig 1A). Genes encoding ntATX-D1 or ntATX-D2 were PCR amplified from the *C. septicum* str. B1 [5,10] DNA using primers listed in Table 1 with 35 cycles of amplification (denaturation at 94˚C for 1min, annealing at 55˚C for 1min, extension at 72˚C for 1min, and a final extension at 72˚C for 7 min). Amplified gene segments were then first cloned between the XhoI and BamHI restriction enzyme sites into the pCR 2.1 TA cloning vector (Invitrogen, Carlsbad, CA) using *E. coli* DH5α host followed by cloning into the pET14b expression vector (Novagen, Madison, WI) in-frame with the N-terminal His-Tag. Proteins were expressed in *E. coli* BL21-DE3 host, as per the standard cloning protocol (TB055 pET System Manual). The N-terminal 6X histidine-tagged recombinant ntATX-D1 and ntATX-D2 proteins were purified using the ProBond Nickel-Chelating Resin columns (Invitrogen, Carlsbad, CA) and analyzed by SDS-PAGE. Immunoreactivity of these proteins to the anti-Histidine antibodies (Invitrogen, Carlsbad, CA) was confirmed by Western blot analysis [17]. In some instances, the yield of soluble cytosolic recombinant proteins was low due to the formation of inclusion bodies containing insoluble protein aggregates. Larger amounts of purified proteins were isolated from inclusion bodies using the Thermo Scientific Inclusion Body Solubilization reagent (Pierce Protein Biology | Thermo Fisher Scientific, Rockford, IL) and purity of the inclusion bodies was checked by SDS-PAGE before proteins were refolded via dialysis using Slide-A-Lyzer Cassette (ThermoFisherScientific). The dialyzed solution was concentrated using Amicon Ultra-15 Centrifugal filter units (Sigma-Aldrich) and the recombinant proteins were then purified with HisPur Ni-NTA columns (Thermo Scientific). Protein purity and the immunoreactivity of recombinant ntATX-D1 and ntATX-D2 proteins was confirmed by gel electrophoresis and Western blot analysis. In addition to the reactivity against anti-Histidine antibodies, the purified ntATX-D1 and ntATX-D2 proteins were also probed with serum from CD-affected turkeys collected in a previous study [5] by Western blot analysis. The primary serum antibody dilution of 1:100 followed by HRP-conjugated anti-turkey IgY antibody (Southern Biotech, Birmingham, AL) at a 1:500 dilution was used in immunoreactivity analysis.

To determine if the purified ntATX-D1 and ntATX-D2 were devoid of toxicity, the hemolytic activity of these proteins was evaluated [18] using 5% defibrinated sheep blood agar plates. Briefly, about 5 mm diameter wells were punched into the agar followed by sealing the bottom with a thin layer of 1% agar. Wells were loaded with 30 μL of purified proteins (10 μg/well) and incubated at 37˚C overnight. The absence of a zone of hemolysis indicated the non-

hemolytic activity of the proteins. As a positive control for hemolysis, the concentrated (20x) *C. septicum* supernatant or the purified *C. perfringens* alpha-toxin (Sigma-Aldrich, St. Louis, MO) was used.

## Immunization, experimental challenge and sampling

Animal experimental protocols used in this research were approved by the North Carolina State University Institutional Animal Care and Use Committee (IACUC protocol 22-181-A). Six-week-old male turkeys were procured from Butterball Farms LLC. (Goldsboro, NC) and were placed on fresh litter consisting of wood shavings in the animal rooms at the Laboratory Animal Research (Biosafety Level 2) facility of the College of Veterinary Medicine, North Carolina State University with unlimited access to water and non-medicated grower feed. The birds were individually identified with leg bands and were divided into four experimental groups with two control groups either receiving or not receiving the *C. septicum* challenge. The birds in the 'negative control' (NCx) group did not receive the *C. septicum* challenge (n = 10). However, this group included five birds that were immunized with ntATX-D1 protein, and five birds immunized with ntATX-D2 protein. The birds in the 'positive control' (PCx) group received only the adjuvant but not the vaccine antigen and were challenged with *C. septicum* (n = 13). The birds in the group termed as 'D1' were immunized with ntATX-D1 protein and challenged with *C. septicum* (n = 15), while those in the group termed as 'D2' were immunized with ntATX-D2 protein and challenged with *C. septicum* (n = 15). Immunization consisted of 100μg of ntATX-D1 or ntATX-D2 protein antigen given via subcutaneous route in the inguinal skin fold on weeks 7, 8 and 9 of age along with the AddaVax (InvivoGen, San Diego, CA), an oil-in-water nano-emulsion vaccine adjuvant equivalent to MF59 at a ratio of 1:1 in a 0.2 mL volume. The NCx group birds were housed separately from the rest of the groups, while the PCx, D1 and D2 groups were housed in one room, but separated by empty floor pens. The NCx group served as the control to evaluate the effect of *C. septicum* challenge in other groups as well as to determine immunization-related adverse effects, if any, while the PCx group provided a control to measure the protective efficacy ntATX-D1 or ntATX-D2 immunizations.

For experimental infection/ challenge at two weeks following to the last immunization, *C. septicum* Str. B1 isolated from commercial turkeys affected with CD [5], and evaluated recently in an experimental challenge model in turkeys and found to be highly virulent [10] was used. *C. septicum* was cultured in Reinforced Clostridial medium (Oxoid, Lenexa, KS) anaerobically at 37˚C for a period of 36–48 hours. Each bird in the infected group was given a dose of 2 mL (0.5 x 10^8 CFU/mL) subcutaneously in the lower pectoral region with 1 mL on each side, while the unchallenged birds received growth medium-only. Clinical signs and mortality were monitored for a period of 72 hours and birds that were found very sick were humanely euthanized followed by necropsy evaluation and tissue collection. All the surviving birds at the end of 72 hours were euthanized using Carbon dioxide and the gross lesions during the necropsy examination were scored as described below. Tissue samples, namely the skin, skeletal muscle and spleen for histology, and the skin, skeletal muscle, spleen and CT for immune gene expression, were collected from birds that were euthanized during the post-challenge observation period (0 to 72 hours) and those that survived till the termination of study (end of 72 hours). The peripheral blood for immunophenotyping was collected from birds at termination prior to necropsy, while for serology, the blood was collected at pre-challenge timepoint when the turkeys were 11-week-old.

## Pathology

At necropsy, gross lesions characteristic of CD in the skin, muscle and spleen were examined and scored as 0 = no lesions; 1 = Dark red/purple-green discoloration, blisters on skin;

2 = Cellulitis (Grade 1), characterized by moist or weepy skin with subcutaneous edema in the breast/ thigh (lower abdomen) and lesions limited to smaller areas (1-3cm) showing exudative hemorrhagic foci; 3 = Cellulitis (Grade 2), characterized by larger areas (>3cm) of skin and subcutaneous lesions showing exudate accumulation accompanied by hemorrhagic and emphysematous/ gas-filled (crepitus) areas; 4 = Subcutaneous lesions extended to muscle with pale red or tan areas of discoloration and multifocal to coalescing hemorrhage and necrosis; 5 = Septicemia: Systemic organ involvement of spleen, liver, heart or other organs showing organomegaly with necrotic foci (extensive multifocal tan foci, and hemorrhage).

For histopathology, skin, skeletal muscle and spleen samples from grossly affected areas, if present, were examined from birds found moribund and euthanized during the clinical monitoring period of the study and of those necropsied at termination. Tissues were fixed in 10% neutral buffered formalin for a minimum of 24 h, processed, sectioned at 5 μm, and stained with hematoxylin and eosin. Sections were examined and lesions were blind-scored by board certified veterinary pathologists, and select sections were additionally stained with Gram stain to identify Gram-positive *Clostridium* rods. Briefly, changes in the skin and skeletal muscle were evaluated based on the criteria of inflammation (including fibrin or edema, heterophils, and lymphocytic or histocytic lesions), gangrenous dermatitis (including lesions of fibrin or edema, cell lysis, and bacteria) and caseous necrosis or granuloma lesions, while splenic changes were evaluated based on the necrosis, cell lysis, and splenic lymphoid depletion and hyperplasia. Each lesion was scored as 0 = no lesion; 1 = focal lesion and limited with <5% of sectioned tissue involved; 2 = multifocal lesion scattered with 5–25% of tissue involved; 3 = multifocal lesion extensive with 30–70% tissue involved; 5 = diffuses lesions with >75% of tissue involved.

## Quantitative real-time PCR

Skin, muscle, spleen and CT were collected in RNAlater solution (Invitrogen, Carlsbad, CA) from turkeys in all the groups (n = 8–10) and stored at −80˚ C until processing. Total RNA was extracted using a Bead Ruptor Elite Bead Mill Homogenizer (OMNI International, Kennesaw, GA) using 1.4 mm Ceramic Beads (OMNI International, Kennesaw, GA) suspended in TRIzol reagent (Invitrogen, Carlsbad, CA) according to the manufacturer's protocol before being treated with a DNA-free Kit (Invitrogen, Carlsbad, CA). Subsequently, cDNA synthesis was performed with 500–1000 ng of purified RNA using a High-Capacity RNA-to-cDNA kit (Applied Biosystems, Waltham, MA) according to the manufacturer's recommended protocol. The resulting cDNA was diluted at 1:10 ratio in nuclease-free water for real-time PCR analysis.

Quantitative real-time PCR using SYBR Green was performed on diluted cDNA using a QuantStudio 6 Flex System and QuantStudio Real-Time PCR Software (Applied Biosystems, Waltham, MA) to quantitate the cellular expression of TLR21, IL-1β, IL-6, IFNγ, IL-4, IL-10 and TGF- β. Briefly, each reaction involved a pre-incubation period of 50˚C for 2 min followed by 95˚C for 2 min, followed by 35–45 cycles of 95˚C for 10 s, 55–64˚C for 5 s, depending on the primers binding suitability, and the elongation step was 72˚C for 10 s. Subsequent melt curve analysis was performed by heating to 95˚C for 15 s, cooling to 60˚C for 1 min, and heating to 95˚C for 15 s. Primers for the amplification of all genes were synthesized by Integrated DNA Technologies (Coralville, IA), and the primer sequences are given in Table 1. Relative expression levels of all target genes were calculated relative to the reference gene β-actin [19].

## Flow cytometry

Blood samples (2 mL) were collected from birds in each group (n = 5–7) and diluted 1:1 with Hank's Balanced Salt Solution (HBSS). Four mL of diluted blood was carefully layered on top

**Table 2. Staining panels used in immunophenotyping.**

| | Marker | Conjugate |
|---|---|---|
| **Panel 1** | CD4 | PE |
| | CD8 | Pacific Blue |
| | CD28 | FITC |
| | IgM | APC |
| | Live/Dead viable dye | APC-Cy7 |
| **Panel 2** | | |
| | CD4 | Pacific Blue |
| | CD8 | APC |
| | CD44 | FITC |
| | MHC II | PE |
| | Live/Dead viable dye | APC-Cy7 |

of Histopaque-1077 (Sigma Aldrich, St. Louis, MO) in a 15 mL conical tube and then centrifuged at 400 x g for 30 min at room temperature with breaks off. PBMCs from the interphase layer were carefully aspirated into a new tube and washed twice with HBSS at 1650 rpm for 10 min at 4˚C. Then the pellet was resuspended in 1mL HBSS. Following the cell count, the suspension was adjusted to 1x 107 cells/mL and 100 μL of which was used for staining, as described previously [20]. Briefly, cells were plated on 96 well round-bottom plates with each well containing 106 cells in 100 μL FACS buffer (PBS with 1% BSA). Primary antibodies were added to each well (0.5–1 μg/106cells). Cells were stained in two different panels of antibody staining due to the paucity of antibody reagents available in multi-color formats (Table 2). All monoclonal anti-chicken antibodies with reported cross-reactivity to turkey cell markers were purchased from Southern Biotech Inc. (Birmingham, AL), which were of mouse origin, and their respective clones are given the parenthesis below. The first staining panel used antibodies against CD4 (CT-4), CD8 (CT-8 recognizing CD8α chain), CD28 (AV-7), and IgM (M-1), and the second panel used anti-CD4 (CT- 4), CD8 (CT-8 recognizing CD8α chain), CD44 (AV-6), and MHC-II (2G11). In both panels, cell viability dye, Live/Dead near IR (Invitrogen, CA) was used to exclude dead cells. Stained cells were washed and fixed with 4% paraformaldehyde before data acquisition using LSR-II flow cytometer (BD Biosciences). The gating strategy included exclusion of doublet cells through forward and side scatters, height and width followed by gating on live cells. Live cell gating was furthermore used as a backbone population to obtain CD4+ and CD8+ cells, CD4+CD44+ cells, CD8+CD44+ cells, CD4+CD28+ cells, CD8+CD28+ cells, and MHC-II+ cells. Single stain and fluorescence minus one control were used for fluorochrome compensation and gating positive cell populations. Data analysis was carried out using the FlowJo software version 10.8.2 (Tree Star, Ashland, OR).

## ELISA

The IgY antibody titers were determined by an indirect ELISA assay [21]. Microtiter plates were coated with the purified recombinant ntATX-D1 or ntATX-D2 antigens at 2.5 μg/mL in 0.1 M carbonate buffer, pH 9.6 overnight at 4˚C. After the coated plates were blocked for 60 min at 37˚C with PBS containing 1% BSA (Sigma, St. Louis, MI), sera (n = 8) from the immunized birds were added to the wells in doubling dilutions starting at 1:200 dilution of the whole sera samples and incubated for 2 h with coated plates at room temperature. After the plates were washed with PBS with 0.1% Tween 20 (PBST), horseradish peroxidase (HRP)–conjugated goat anti-turkey IgY (heavy and light chains; Southern Biotech, Birmingham, AL) diluted to 1:10000 in PBST, 1% BSA was added to the microplates, and the mixture was incubated for 60 min at room temperature.

After five repeated washing of the plates with PBST using a Bio-Tek plate washer (Winooski, VT), the color reaction was developed by using the HRP substrate solution (Pierce TMB substrate kit, Waltham, MA), following the manufacturer's instructions. The reaction was stopped by adding 50 µl of the stop solution (0.16 M Sulfuric acid) to each well and the absorbance was measured at an optical density (OD) of 450 nm using a Bio-Tek microplate reader.

### Statistical analysis

All the data were analyzed using GraphPad Prism V9.5.1 (GraphPad software, San Diego, CA, USA). The mortality data were analyzed by simple survival analysis (Kaplan-Meier) and Mantel-Cox as well as Gehan-Breslow-Wilcoxon tests, while the data related to gross pathology and histopathology were analyzed by the non-parametric Kruskal-Wallis test and the values were expressed as the median score. The gene expression and cellular data were first tested for normal distribution (Shapiro-Wilk test) followed by one-way ANOVA (parametric or non-parametric) test analyses. The normally distributed data were analyzed by Tukey's multiple comparison test, while the data not normally distributed were analyzed by the Kruskal-Wallis test followed by Dunn's multiple comparison test. For ELISA data, the average OD values of immunized and control samples obtained in each of the dilutions were compared using the two-way ANOVA method applying Sidak's pairwise multiple comparison test. The asterisks above the median range or standard error of mean bars denoted in the graphs indicate statistical significance: $^*P < 0.05$, $^{**}P < 0.01$ or $^{***}P < 0.001$.

## Results

### Expression and purification of recombinant proteins

Genes encoding ntATX-D1 or ntATX-D2 regions (Fig 1A) were amplified from *C. septicum* (Fig 1B) and cloned into the 'cloning' vector (pCR2.1-TA) followed by expression using the vector, pET14b, as histidine-tagged recombinant ntATX-D1 (~45 kDa) or ntATX-D2 (~25 kDa) proteins. The purified recombinant proteins were confirmed by immunoreactivity with anti-His antibodies in Western blot assays, along with the empty vectored *E. coli* clones as the negative control, NCx (Fig 1C). Furthermore, serum collected from turkeys infected with *C. septicum* was also used in Western blot analysis and found that both ntATX-D1 and ntATX-D2 purified proteins showed immunoreactivity, as shown in Fig 1D. The original images of gels and blots used in preparing Fig 1 are given in S1 Fig. Furthermore, the hemolytic assay showed that the purified proteins were devoid of toxicity (S2 Fig).

### Protection assessment

Protection against *C. septicum* challenge was assessed by monitoring mortality and clinical signs in birds post-challenge and evaluating gross lesions and histopathological changes in both local (skin and muscle) and systemic (spleen) tissues. As shown in Fig 2A, mortality was observed in the PCx group as early as by 24 hours culminating in a total of 46.2% (6 out of 13 birds) deaths by the end of 72 hours. Conversely, the onset of mortality in D1- or D2-immunized birds was delayed and the total mortality by 72 hours was also significantly reduced to 13.3% (2 out of 15 birds). Gross pathology scores showed that all the challenged groups developed lesions when compared to the uninfected NCx birds, while the D2-immunized turkeys had significantly reduced lesions than the PCx group, supporting protection against experimental CD infection (Fig 2B). It was noteworthy that the total number of birds with lesions scored ≤ 2 in the D2-immunized group was 53.3% (8 of 15), when compared to PCx group that had 7% (1 of 13) of birds with lesions scored ≤ 2, indicating that the D2-immunized birds

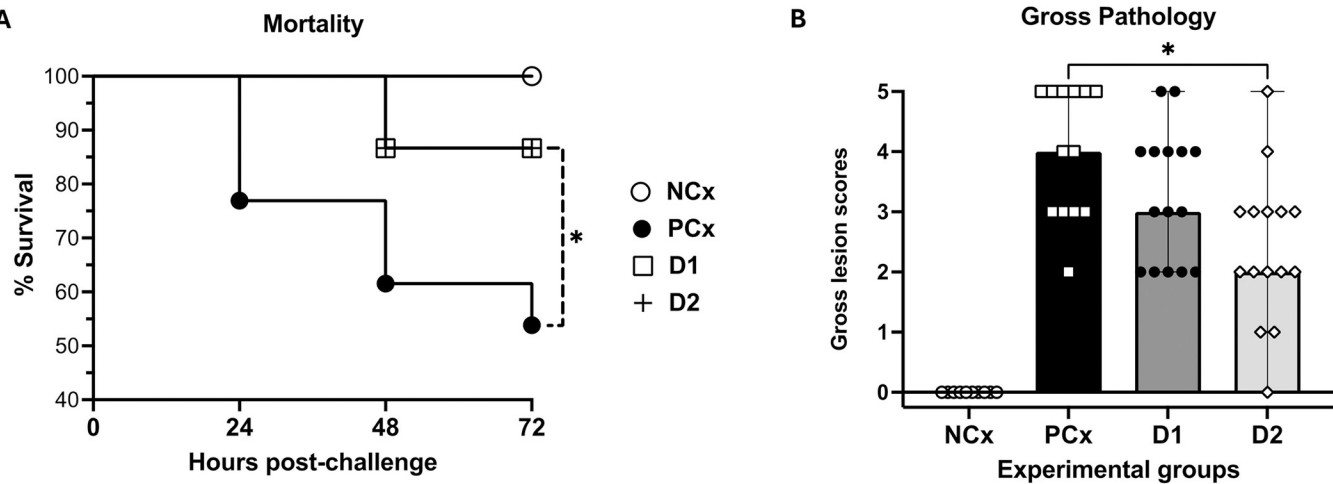

**Fig 2. Mortality and gross pathology lesions in turkeys following *C. septicum* challenge.** Turkeys were immunized with subunit recombinant proteins at 7, 8 and 9 weeks of age followed by a *C. septicum* challenge at two weeks post-last immunization. Mortality was monitored for a period of 72 hours as shown in the % survival chart (panel A), and the gross lesions were scored (panel B), as described in the materials and methods section. NCx- negative control, unchallenged (n = 10); PCx- positive control, unimmunized and challenged (n = 13); D1- immunized with ntATX-D1 protein and challenged (n = 15); D2- immunized with ntATX-D2 protein and challenged (n = 15). Bars in the graph showing gross lesion scores represent median score with range and the asterisks indicate statistical significance, when compared to PCx group: * $P < 0.05$.

had less progression of CD severity. Although the D1-immunization resulted in 33.33% (5 of 15) of birds with a score of $\leq 2$, this group had 7 birds that developed severe lesions, scored $\geq 4$, supporting a lack of statistically significant protection against CD in this group.

As shown in Fig 3, the histopathological changes in the skin and muscle tissues were evaluated based on the criteria of inflammation, gangrenous dermatitis and caseous necrosis/granuloma lesions, while the splenic changes were evaluated based on the necrosis, cell lysis, and lymphoid depletion and hyperplasia. The PCx and D1-immunized groups had significantly higher scores for skin lesions characterized under the categories of inflammation and gangrenous dermatitis when compared to NCx group (Fig 3A). However, no significant difference was observed between the D2 and NCx groups. Furthermore, no significant changes were observed in the skin caseous necrosis/granuloma lesion category between the groups. The skeletal muscle tissue examination revealed that the PCx group had significantly higher scores of lesions indicative of inflammation, gangrenous dermatitis and myopathy when compared to NCx birds (Fig 3B). Additionally, the D1-immunized birds also had significantly higher scores related to inflammation and gangrenous dermatitis lesions in comparison to NCx group of birds. However, the scores of lesions characteristic of gangrenous dermatitis and myopathy in the D2-immunized birds were significantly lower in comparison to those in the PCx group. No significant changes were observed in the muscle caseous necrosis/granuloma lesion category between the groups. Furthermore, no significant changes were observed in the splenic necrosis, cell lysis, or lymphoid depletion and hyperplasia between the groups (Fig 3C). However, the PCx group had a numerically increased score for the lesions characterized by splenic lymphoid depletion and hyperplasia. Photomicrographs representing lesions scores in the skin, muscle and spleen are shown in S3–S5 Figs, respectively.

## Immune gene expression

Expression of TLR21, IL-1β, IL-6, IFNγ, IL-4, IL-10 and TGF-β genes in the skin, muscle, cecal tonsil and spleen tissues from the D1- and D2-immunized groups along with the NCx and PCx controls was determined.

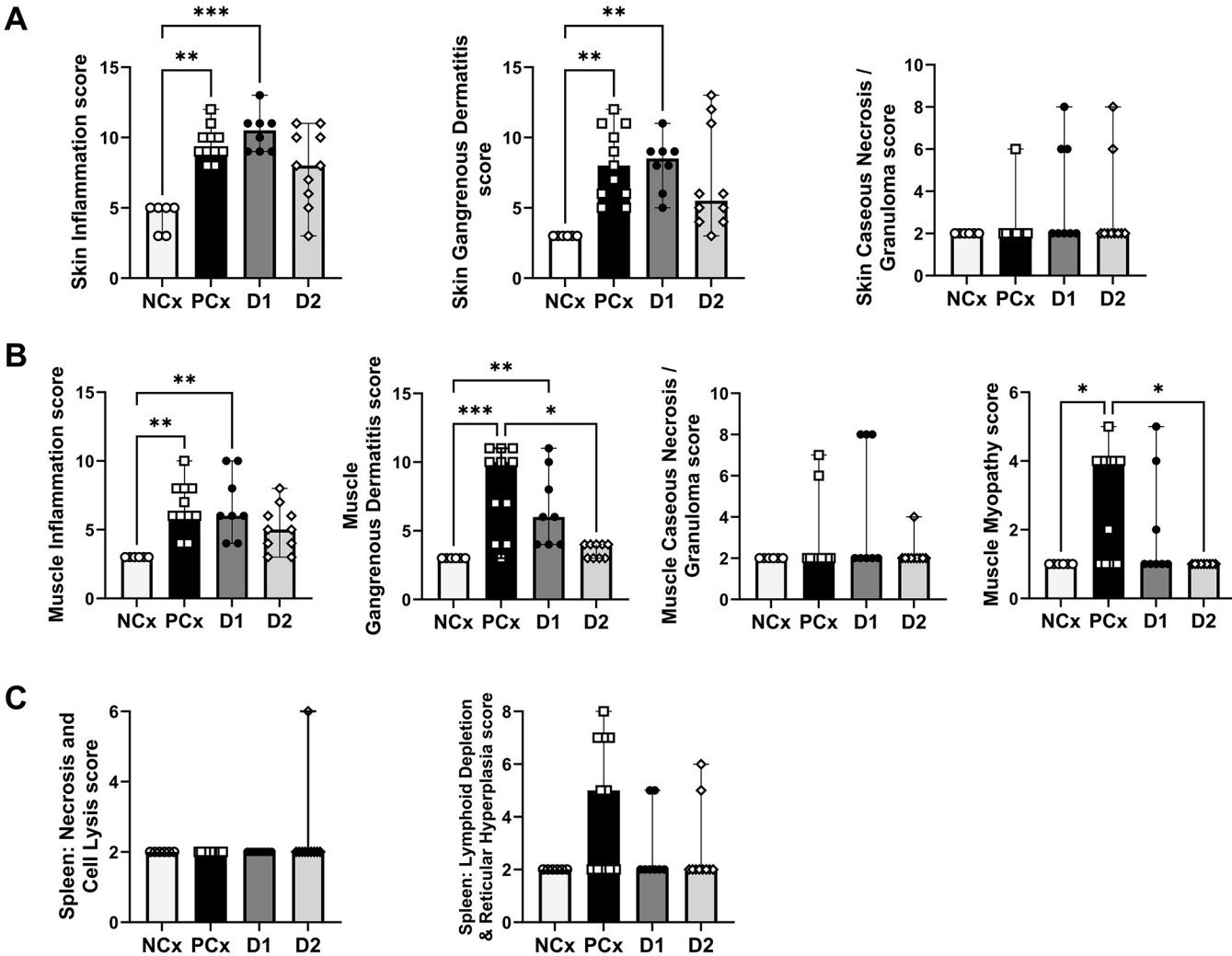

**Fig 3. Histopathological changes in immunized turkeys following *C. septicum* challenge.** Panel A represents changes in the skin indicating lesion scores evaluated based on the criteria of inflammation (including fibrin/edema, heterophils, and lymphocytic/ histocytic lesions), gangrenous dermatitis (including lesions of fibrin/ edema, cell lysis, and bacteria) and caseous necrosis/ granuloma lesions. Panel B represents changes in the muscle indicating lesion scores evaluated based on the criteria of inflammation (including fibrin/ edema, heterophils, and lymphocytes/ histocyte lesions), gangrenous dermatitis (including lesion scores for fibrin/ edema, cell lysis, and bacteria), caseous necrosis and granuloma lesions, and muscle myopathy lesions. Panel C represents changes in the spleen indicating lesion scores evaluated based on the criteria of splenic necrosis, cell lysis, and splenic lymphoid depletion and hyperplasia. NCx- negative control, unchallenged (n = 6); PCx- positive control, unimmunized and challenged (n = 11); D1- immunized with ntATX-D1 protein and challenged (n = 8); D2- immunized with ntATX-D2 protein and challenged (n = 10). Bars represent median score with range and the asterisks above the bars indicate statistical significance (* $P < 0.05$, ** $P < 0.01$ or *** $P < 0.001$) between the groups.

In the skin (Fig 4), transcription of IL-1β in the PCx birds was significantly increased when compared to NCx group, while D2-immunized birds had significantly higher expression of IL-6 and IL-10 genes when compared to NCx birds. Additionally, the IL-10 transcription in D2-immunized birds was also significantly higher than those in the PCx group. Furthermore, D2-immunized birds had significantly increased TLR21 gene expression when compared to NCx birds. No significant changes in the expression of IFNγ, IL-4 or TGF- β genes between groups was observed.

In the muscle (Fig 5), the expression of IL-1β and IL-6 genes in the PCx and D1 groups was significantly increased when compared to NCx group, while the D2-immunized birds had significantly downregulated expression of IL-1β and IL-6 genes than the PCx birds. Furthermore,

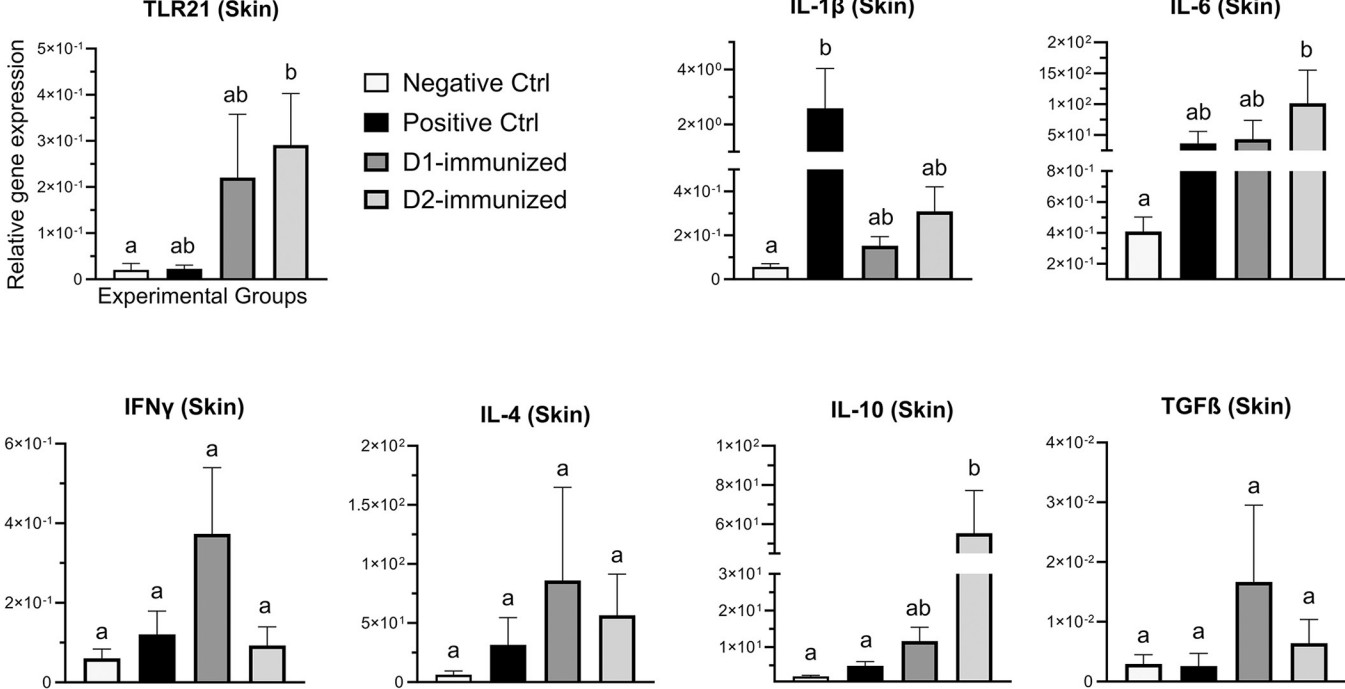

**Fig 4. Immune gene expression in skin tissue of immunized turkeys.** Turkeys were immunized with subunit recombinant proteins at 7, 8 and 9 weeks of age followed by a *C. septicum* challenge at two weeks post-last immunization. Tissue samples from birds (n = 8) in the experimental groups were collected at termination for RNA extraction and cDNA synthesis. Real-time PCR to quantify the expression of genes indicated in the figure was performed and the expression levels are shown as relative to β-actin reference gene. Negative Ctrl- Control, unchallenged; Positive Ctrl- Control, unimmunized and challenged; D1- immunized with ntATX-D1 protein and challenged; D2- immunized with ntATX-D2 protein and challenged. Different letters above the standard error of mean bars indicate significant difference (P<0.05) between the groups.

the transcription of IFNγ in the D2-immunized group was also significantly downregulated than the PCx group. No significant changes were observed in the expression of TLR21, IL-4, IL-10 or TGF-β genes between the groups.

In CT (Fig 6), the transcription of IL-1β and IFNγ genes in PCx group was significantly higher compared to birds in the NCx, while the expression of IL-1β and IL-6 genes in D2-immunized birds was significantly downregulated than those in the PCx group. The expression of IL-6 and TLR21 genes in D2-immunized group was also significantly downregulated compared to D1-immunized birds. Additionally, the transcription of IL-10 gene was significantly higher in D1-immunized birds than the NCx group. However, no significant changes observed in the expression of IL- 4 and TGF-β genes between the groups.

In spleen (Fig 7), IL- 4 transcription in PCx group was significantly higher than the NCx group, while the expression of IL- 6 and IL- 4 genes in the D2 immunized birds was significantly downregulated compared to PCx group. No significant changes were observed in the transcription of IL-1β, IFNγ TLR21, IL-10 or TGF-β genes between the treatment groups.

In addition to the figures presented here, two summary tables related to the immune gene expression findings are given. To evaluate the effect of *C. septicum* challenge, the Table 3 summarizes the significant (*P* < 0.05) changes in the treatment groups (PCx, D1 and D2) compared to NCx group, while to determine the immunization-induced changes, the Table 4 summarizes the significant (*P* < 0.05) changes in the D1- or D2-immunized groups compared to PCx group is given in Table 4. The tables based the statistical method of multiple comparisons between the groups (Tukey's or Kruskal-Wallis tests) used in denoting significance (*P* < 0.05) shown in Fig 4 through Fig 7.

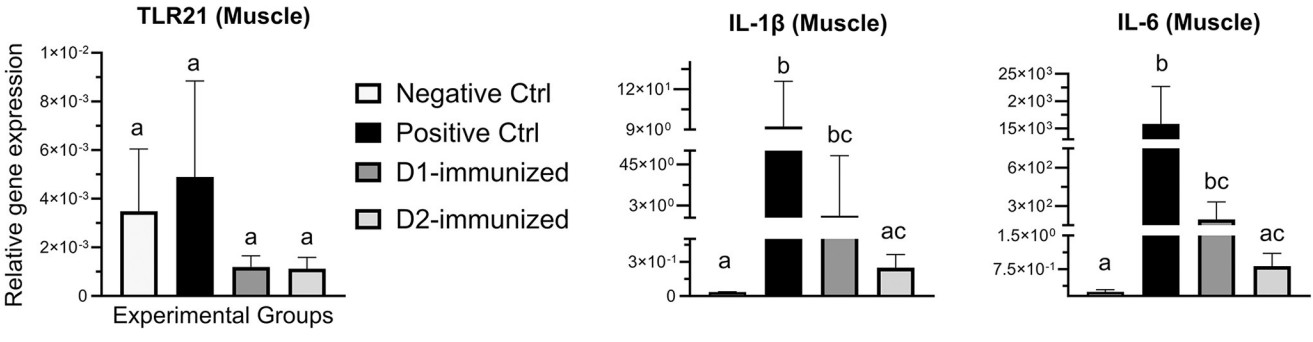

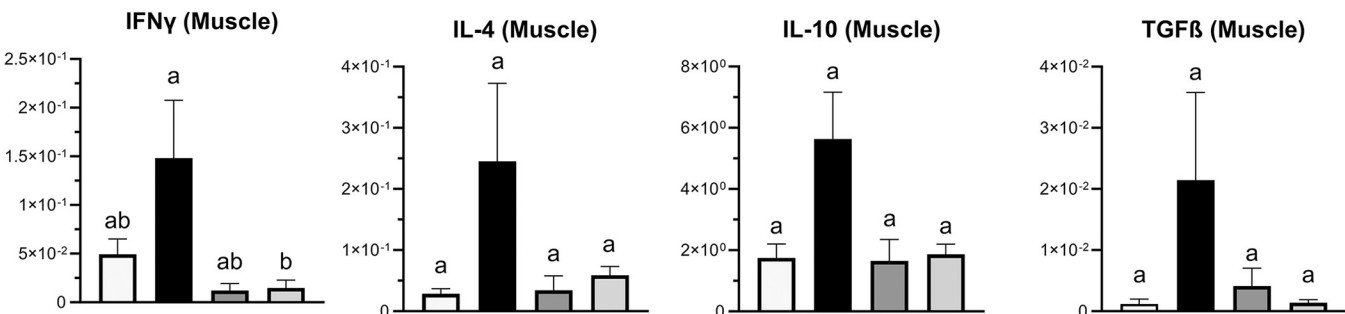

**Fig 5. Immune gene expression in muscle tissue of immunized turkeys.** Turkeys were immunized with subunit recombinant proteins at 7, 8 and 9 weeks of age followed by a *C. septicum* challenge at two weeks post-last immunization. Tissue samples from birds (n = 8) in the experimental groups were collected at termination for RNA extraction and cDNA synthesis. Real-time PCR to quantify the expression of genes indicated in the figure was performed and the expression levels are shown as relative to β-actin reference gene. Negative Ctrl- Control, unchallenged; Positive Ctrl- Control, unimmunized and challenged; D1- immunized with ntATX-D1 protein and challenged; D2- immunized with ntATX-D2 protein and challenged. Different letters above the standard error of mean bars indicate significant difference (P<0.05) between the groups.

## Immunophenotyping

To evaluate the immunization-induced cellular responses, the frequencies of CD4+, CD4+CD28+ and CD4+CD44+, CD8+, CD8+CD28+ and CD8+CD44+, as well as MHC-II+ cell populations in the PBMCs were analyzed using flow cytometry. Due to the unavailability of turkey-specific antibody reagents for use in flow cytometry staining, anti-chicken antibodies with reported cross-reactivity to turkey cell markers were used [10]. Although the anti-IgM antibody was included in one of the two staining panels, its binding to turkey cells in obtaining a distinct positive cell population was unsuccessful; hence, it was excluded from the present study analysis. The gating strategy used to allow an accurate analysis of these cell types is shown in Fig 8A. It is noteworthy that the lack of anti-chicken CD3 cross-reactivity with turkey CD3 led us to using the 'live' cells as the backbone population for gating the CD4+/CD8+ cells. The frequency of CD4+ cells (Fig 8B), CD4+CD44+ cells (Fig 8C) and CD4+CD28+ cells (Fig 8E) were significantly decreased in PCx birds compared to NCx group. However, the D2-immunized birds were found to have significantly increased populations of CD4+ cells (Fig 8B), and CD4+CD28+ cells (Fig 8E) when compared to PCx group. No significant changes in the CD4+ cells (Fig 8B) and CD4+CD44+ (Fig 8C) frequencies were observed in D1 and D2 group when compared with the NCx group. The frequency of CD8+ cells (Fig 8B), CD8+CD44+ cells (Fig 8D) and CD8+CD28+ cells (Fig 8F) cells were significantly increased in D1and D2 immunized birds than the PCx group. Additionally, the birds in the PCx group were found to have significantly decreased populations of CD8+CD28+ cells (Fig 8F) in

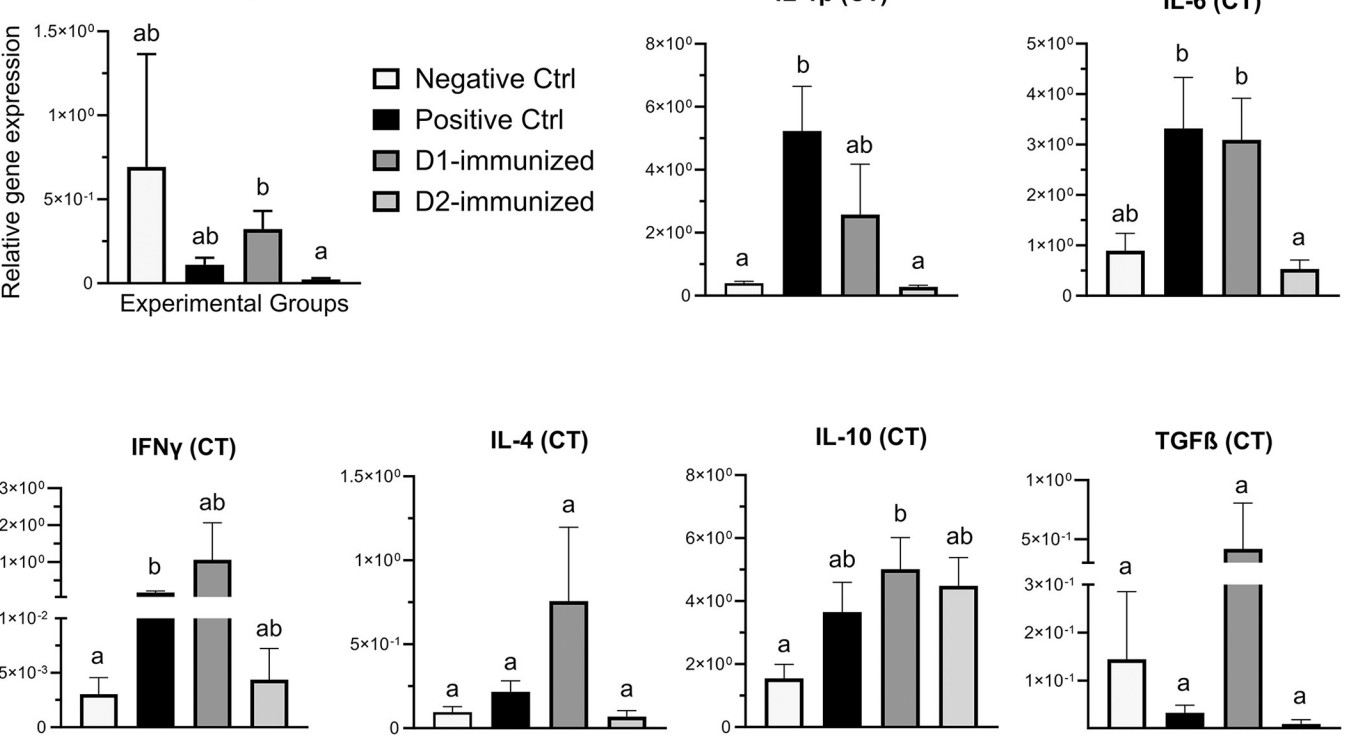

**Fig 6. Immune gene expression in cecal tonsil tissue of immunized turkeys.** Turkeys were immunized with subunit recombinant proteins at 7, 8 and 9 weeks of age followed by a *C. septicum* challenge at two weeks post-last immunization. Tissue samples from birds (n = 8) in the experimental groups were collected at termination for RNA extraction and cDNA synthesis. Real-time PCR to quantify the expression of genes indicated in the figure was performed and the expression levels are shown as relative to β-actin reference gene. CT- Cecal tonsil; Negative Ctrl- Control, unchallenged; Positive Ctrl- Control, unimmunized and challenged; D1- immunized with ntATX-D1 protein and challenged; D2- immunized with ntATX-D2 protein and challenged. Different letters above the standard error of mean bars indicate significant difference (P<0.05) between the groups.

comparison to those in the NCx group. Furthermore, the frequency of MHC-II+ cells was significantly higher in D2-immunized group when compared to the NCx and PCx groups (Fig 8G).

## Antibody evaluation

The serum samples, collected prior to the *C. septicum* challenge, from immunized turkeys were evaluated to determine antigen (ntATX-D1 or ntATX-D2)-specific IgY levels. As shown in Fig 9, the immunized birds showed a significant ($P < 0.05$) increase in the antigen-specific IgY antibody levels when compared to control, as determined by their OD values over the course of 9-point serum doubling dilutions.

## Discussion

In recent years, Clostridial diseases are negatively impacting the poultry health and economy. One such disease affecting the poultry industry, particularly the turkey sector, is *C. septicum*-induced Clostridial dermatitis. Although the therapeutic antibiotics can control CD, their availability and the effectiveness coupled with the increasing risk of antimicrobial resistance warrant immediate antibiotic alternative strategies such as vaccines. To this end, the present study focused on *C. septicum* alpha toxin (ATX) to identify two non-toxic domains, namely the ntATX-D1 and ntATX-D2, and develop subunit protein-based vaccines for immunizing

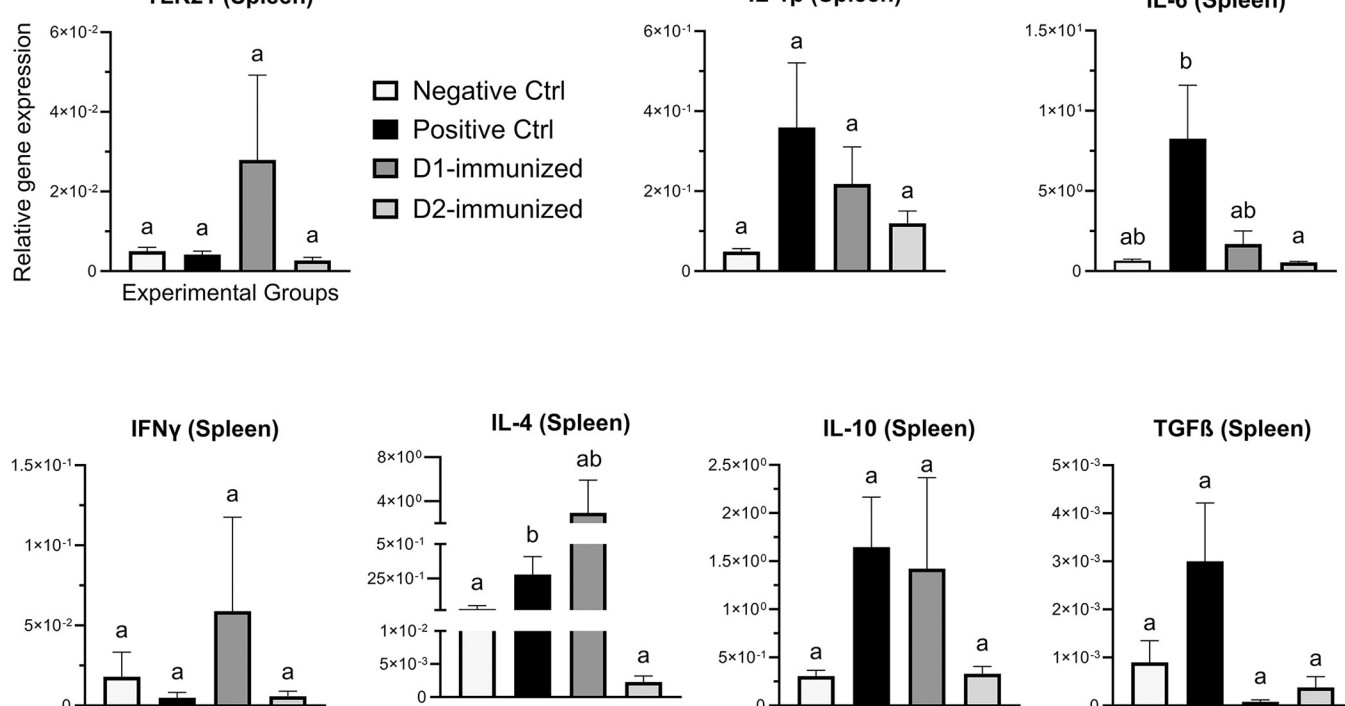

**Fig 7. Immune gene expression in spleen tissue of immunized turkeys.** Turkeys were immunized with subunit recombinant proteins at 7, 8 and 9 weeks of age followed by a *C. septicum* challenge at two weeks post-last immunization. Tissue samples from birds (n = 8) in the experimental groups were collected at termination for RNA extraction and cDNA synthesis. Real-time PCR to quantify the expression of genes indicated in the figure was performed and the expression levels are shown as relative to β-actin reference gene. Negative Ctrl- Control, unchallenged; Positive Ctrl- Control, unimmunized and challenged; D1- immunized with ntATX-D1 protein and challenged; D2- immunized with ntATX-D2 protein and challenged. Different letters above the standard error of mean bars indicate significant difference (P<0.05) between the groups.

turkeys against CD. The results found that while both proteins were able to significantly lower mortality, the ntATX-D2 provided a more robust protective immunity, which was supported by gross and histopathological findings as well as immunological parameters.

*Clostridium septicum* is an anaerobic bacterial pathogen, which produces ATX, a cytolytic necrotizing toxin implicated as the key virulence factor in the pathogenesis of CD [8,9]. We recently reported that *C. septicum* induces a robust inflammatory response in local as well as

**Table 3. Summary of changes in the expression of immune genes in treatment groups compared to unchallenged control (NCx) group.**

| Genes | Skin | | | Muscle | | | Cecal tonsil | | | Spleen | | |
|---|---|---|---|---|---|---|---|---|---|---|---|---|
| | PCx | D1 | D2 | PCx | D1 | D2 | PCx | D1 | D2 | PCx | D1 | D2 |
| TLR21 | = | = | + | = | = | = | = | = | = | = | = | = |
| IL-1β | + | = | = | + | + | = | + | = | = | = | = | = |
| IL-6 | = | = | + | + | + | = | = | = | = | = | = | = |
| IFNγ | = | = | = | = | = | = | + | = | = | = | = | = |
| IL-4 | = | = | = | = | = | = | = | = | = | + | = | = |
| IL-10 | = | = | + | = | = | = | = | + | = | = | = | = |

+ denotes increased (*P* < 0.05) and–for decreased (*P* < 0.05), while = indicates no significant changes in the target gene expression in groups when compared to unchallenged negative control (NCx). PCx- positive control (unimmunized and challenged), D1-immunized with ntATX-D1 antigen and challenged and D2-immunized with ntATX-D1 antigen and challenged.

**Table 4. Summary of changes in the expression of immune genes in immunized groups compared to unimmunized and challenged (PCx) group.**

| Genes | Skin | | Muscle | | Cecal tonsil | | Spleen | |
|---|---|---|---|---|---|---|---|---|
| | D1 | D2 | D1 | D2 | D1 | D2 | D1 | D2 |
| TLR21 | = | = | = | = | = | = | = | = |
| IL-1β | = | = | = | - | = | - | = | = |
| IL-6 | = | = | = | - | = | - | = | - |
| IFNγ | = | = | = | - | = | = | = | = |
| IL-4 | = | = | = | = | = | = | = | - |
| IL-10 | = | + | = | = | = | = | = | = |

+ denotes increased ($P < 0.05$) and–for decreased ($P < 0.05$), while = indicates no significant changes in the target gene expression in groups when compared to unimmunized and challenged positive control (PCx). D1-immunized with ntATX-D1 antigen and challenged and D2-immunized with ntATX-D1 antigen and challenged.

systemic tissues during CD in turkeys [5,10], and much of it is attributable to the ATX-mediated damage [2,14]. The present study found that immunization of turkeys with ntATX protein antigens, specifically the ntATX-D2, can lower mortality and reduce CD-induced pathology, suggesting that the ATX seems to play a key role in *C. septicum* virulence.

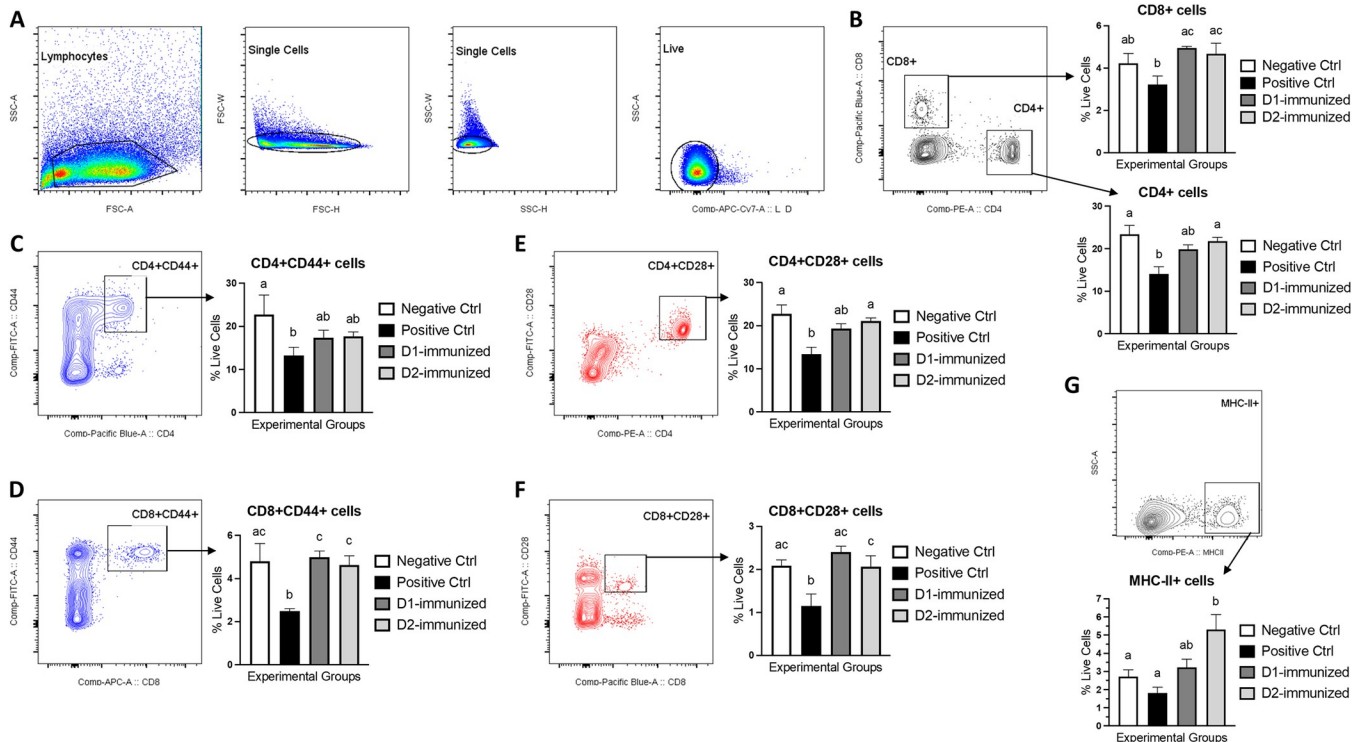

**Fig 8. Peripheral blood cellular response in immunized turkeys.** Turkeys were immunized with subunit recombinant proteins at 7, 8 and 9 weeks of age followed by a *C. septicum* challenge at two weeks post-last immunization. Peripheral blood samples from birds (n = 5–7) in the experimental groups were collected at termination for the preparation of PBMC suspensions. Cells were stained with antibodies against CD4, CD8, MHC-II, CD44, and CD28 molecules for flow cytometry-assisted immunophenotyping analysis. The gating strategy included exclusion of doublet cells through forward and side scatters, height and width followed by gating on live cells (panel A). Live cell gating was furthermore used as a backbone population to obtain CD4+ and CD8+ cells (panel B), CD4+CD44+ cells (panel C), CD8+CD44+ cells (panel D), CD4+CD28+ cells (panel E), CD8+CD28+ cells (panel F), and MHC-II+ cells (panel G). Negative Ctrl-Control, unchallenged; Positive Ctrl- Control, unimmunized and challenged; D1- immunized with ntATX-D1 protein and challenged; D2- immunized with ntATX-D2 protein and challenged. Different letters above the standard error of mean bars indicate significant difference (P<0.05) between the groups.

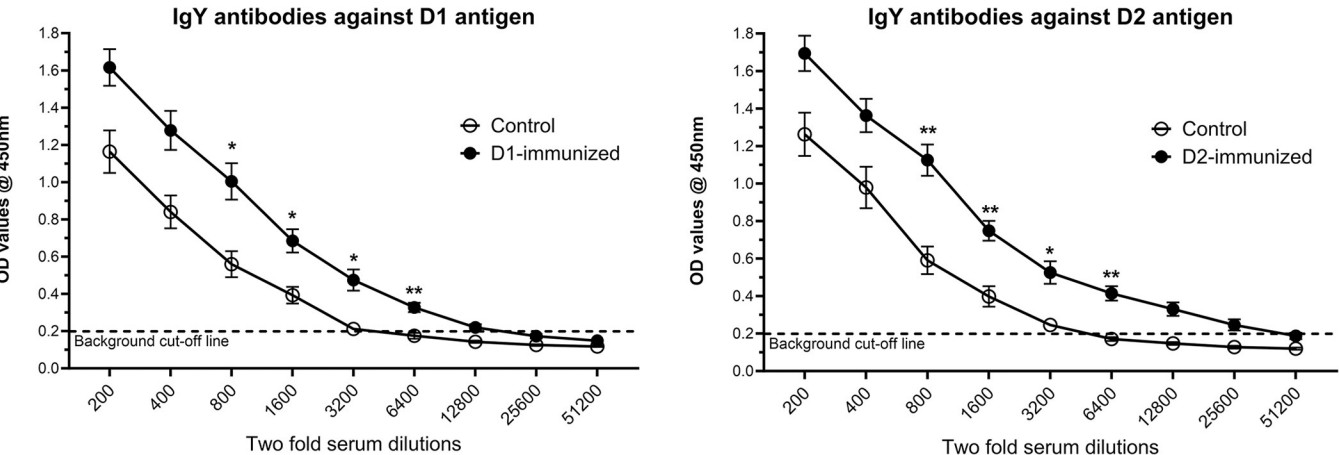

**Fig 9. Serum antibody response in immunized turkeys.** Turkeys were immunized with subunit recombinant proteins at 7, 8 and 9 weeks of age and the serum samples from birds (n = 8) in the experimental groups were collected at two weeks post-last immunization prior to the *C. septicum* challenge. The antigen-specific turkey IgY antibody levels were measured by ELISA using a two-fold serum dilution method. Negative Ctrl- Control; D1- immunized with ntATX-D1 protein; D2- immunized with ntATX-D2 protein. Statistical significance is indicated as * $P < 0.05$ or ** $P < 0.01$.

Furthermore, the ntATX-D1 and ntATX-D2 immunizations leading to an antigen-specific seroconversion indicates that antibodies may play an important role in protection against CD in turkeys. Previous work using the *C. septicum* bacterin toxoid for immunizing turkeys against CD has noted the importance of antibodies against ATX in alleviating CD-induced mortality and disease severity. For example, our previous work showed that subcutaneous immunization of commercial turkeys with *C. septicum* bacterin-toxoid can induce bacterin-specific serum antibodies, as well as reduce CD morbidity and mortality [11]. More recently, *C. septicum* bacterin-toxoid adjuvanted with water-in-oil Montanide emulsion vaccination, via subcutaneous route, in commercial turkeys led to reduced mortality and CD incidences and vaccinated birds were found to develop antibodies [22]. Furthermore, Lancto et.al. (2014) used a noncytolytic *C. septicum* alpha-toxin (NCAT) peptide, constructed by excluding the 28-aminoacid cytolytic domain from the native ATX, with or without *C. septicum* bacterin + alpha-toxoid in a turkey subcutaneous immunization experiment [8]. Authors found that while the NCAT immunization was able to reduce mortality, the birds receiving bacterin + alpha-toxoid showed the highest protective efficacy when compared to other groups. Although these studies infer that ATX offers protective immunogenicity, the present study findings provide specific information related to the ATX domains, specifically the ntATX-D2 region, that play a key role in protection against CD in turkeys. It was also noteworthy that the clinical protection observed in the ntATX-D2 immunized groups was not only supported by the reduced mortality but also by the significant reduction in the gross and histopathology lesion scores. A case in point was the histopathological observation that ntATX-D2 immunized birds had significantly reduced muscle lesions associated with gangrenous dermatitis and myopathy compared to the unimmunized (PCx) group, while no inflammatory changes were observed in the skin of these birds. These findings suggest that ntATX-D2 can offer superior CD protection ability over ntATX-D1 in turkeys, and thus, can be explored further as an efficacious and viable vaccine candidate for use in future vaccine research development and applications.

The immune gene expression analysis revealed two important pieces of information related to CD pathogenesis and vaccine-induced protection. (1) The unimmunized but challenged (PCx group) turkeys had a significantly ($P \leq 0.05$) elevated expression of pro-inflammatory

cytokine (IL-1β, IL-6 and IFNγ) genes in the skin, muscle and CT tissues compared to unchallenged (NCx) birds. This observation suggested that the virulent *C. septicum* can induce a robust local as well as systemic inflammatory response leading to tissue damage, which may chiefly be orchestrated via their secretory toxins, particularly the ATX. In support of this, our previous work found that turkeys infected with *C. septicum* during either a field CD outbreak [5] or an experimental infection [10] had a significantly higher transcription of IL-1β, IFNγ and IL-6 in the local (skin and muscle) and systemic (spleen) tissues when compared to clinically healthy or uninfected birds. In the context of gangrenous dermatitis field outbreaks in chickens, two previous studies also showed increased transcription of pro-inflammatory cytokine and chemokine genes in the skin tissues as well as the presence of higher numbers of K55 +, K1+, CD8+ and MHC class II+ intradermal lymphocytes in affected birds [23,24]. These observations suggest that therapeutic or prophylactic interventions such as vaccines capable of culminating local and/or systemic inflammation can successfully prevent CD progression in turkeys. (2) Turkeys receiving ntATX-D2 immunization, when compared to PCx group, were found to have a significantly reduced expression of pro-inflammatory cytokine (IL-1β, IL-6 and IFNγ) genes in both local and systemic tissues accompanied by a significant increase in the expression of IL-10 and IL-6 cytokine genes in the skin. While IL-10 is known to dampen inflammation [25], the IL-6 is a pleotropic cytokine whose biological functions have also been linked to immunomodulation during the stages of infection resolution [26]. Another intriguing finding was also that the ntATX-D2 immunized birds had significantly higher transcription of CpG-DNA sensing TLR21 receptor, which is known to recognize Clostridial pathogens [5,27,28] and aid in antibody production by B cells during bacterial infections [29,30]. Taken together, although a direct effect of ntATX-D2 immunization on the cytokine-mediated regulation of inflammation cannot be established based on the present study findings, it is reasonable to suggest that immunization-driven antibody and cellular responses leading to an effective infection control may have led to the increased expression of immunoregulatory cytokines [31,32].

Vaccine-induced CD4+ and/or CD8+ T cell responses are important in immunity against bacterial pathogens, including *Clostridium* species [33–35]. The immunophenotyping analysis of PBMCs in the present study made two observations: (1) The *C. septicum* infection in the unimmunized-challenged group (PCx) led to a decreased CD4+ and CD8+ cell frequencies compared to unchallenged (NCx) birds, suggesting a possible ATX-mediated cellular toxicity [16,36]. Consequently, the frequencies of activated CD4+ and CD8+ were also found significantly reduced when compared to NCx group. (2) The immunization with ntATX proteins, specifically the ntATX-D2, could rescue the infection-induced CD4+ and CD8+ cellular depletion since the frequencies of these cells in the immunized turkeys were significantly higher than the PCx, with no changes compared to NCx birds. Furthermore, the CD4+ and CD8+ cellular expression of CD28 and CD44 molecules, which are the markers for activated T cells [37,38] was determined. The findings showed that while the ntATX-D1 immunization could significantly increase CD8+ cell activation (CD44+ and CD28+ cells), the ntATX-D2 immunized birds had significantly higher number of both activated CD4+ (CD28+) and CD8+ (CD44+ and CD28+) cells than the PCx birds, with no changes when compared to NCx group. These observations indicated that the prime-boost immunization-induced enhancement of CD4+ and/or CD8+ effector, and perhaps also the memory, cell responses may have played a key role in protecting turkeys against CD [39]. The more robust protection offered by ntATX-D2 immunization can be explained by its higher induction of CD4+ cell responses since an important feature of activated CD4+ T cells is their help to naïve B cell differentiation into antibody-secreting plasma cells via MHC-II mediated antigen presentation [40]. In support of this, an increase in the frequency of MHC-II+ cells in the ntATX-D1 immunized birds

was observed, indicating a possible enhanced antigen presentation and T cell effector function [33,41]. The observation that ntATX-immunizations showed enhanced CD8+ cell proliferation and activation suggests that these cells may have a role in protective immunity against extracellular pathogens [42], including *C. perfringens* in chickens [20]. This is because *Clostridium* pathogens are shown to survive within the macrophages [43] and that the activated CD8 + T cells can help activate macrophages to clear the infection [33]. It is also noteworthy here that due to the unavailability of anti-turkey CD3 antibody, the present study lacked a T cell-specific marker for staining; however, based on the cellular expression of CD28 and CD44, we can reasonably presume that the CD4+ and CD8+ cells reported here are T cells. More work is certainly needed to elucidate precise T cellular mechanisms of protection in the context of our findings.

In conclusion, the present study showed that (1) Immunization of turkeys against CD, using ATX as a target vaccine antigen, can effectively prevent mortality and reduce disease severity; (2) Of the two non-toxic antigenic domains of ATX, the ntATX-D2 seems to offer the most efficacious vaccine antigen platform in achieving protective immunity against CD in turkeys; and (3) The immune mechanisms of immunization-driven protection may include an effective antibody response along with enhanced CD4+ and CD8+ T cell functions, which collectively can control infection-induced inflammation.

## Supporting information

**S1 Fig. Raw images.** Pic A: Lanes 4 (912bp), 5 (neg ctrl), 6 (ladder) and 7 (531bp) were cropped from the gel pic below for the main Fig 1B. Pic B: Lanes R1, R2 and R3 are the purified ntATX-D1. To make the main Fig 1C, lane R3 was cropped and used along with the protein ladder L. Pic C: Lanes R1 and R2 are the purified ntATX-D2. To make the main Fig 1C, lane R2 was cropped and used along with the protein ladder L. Pic D: Lanes 2 and 3- ntATX-D2, Lanes 4 and 5- ntATX-D1, Lane 1- Ladder. To make the main Fig 1D, the lanes 1, 2 and 4 were cropped.
(TIF)

**S2 Fig. Non-hemolytic activity of ntATX-D1 and ntATX-D2.** The hemolytic activity of purified ntATX-D1 and ntATX-D2 was evaluated using 5% defibrinated sheep blood agar plates. Wells were punched into the agar and loaded with 30 μL of purified proteins (10 μg/well) and incubated at 37˚C overnight. The absence of a zone of hemolysis indicated the non-hemolytic activity of the proteins. As a positive control for hemolysis, the purified *C. perfringens* alpha-toxin was used.
(TIF)

**S3 Fig. Histopathological changes in the skin of turkeys representing experimental groups.** (A) Negative control. Occasional samples had focal, mild, often perivascular lymphocytic and heterophilic inflammation in the superficial dermis (arrowhead), affecting less than 5% of the tissue (lesion scores = 2). Bar = 150 μm. (B) Positive control. The dermis is markedly expanded by fibrin and edema (arrowhead), consistent with a lesion score of 4. Bar = 325 μm. (C) ntATX-D1 vaccinated bird. Skin shows cell lysis of erythrocytes (arrowhead) multifocally affecting 30–75% of the tissue, consistent with a lesion score of 4. Bar = 100 μm. (D) ntATX-D1 vaccinated bird. The deep dermis contains a discrete caseous granuloma characterized by an outer rim of macrophages and heterophils surrounding an inner, hypereosinophilic core of caseous necrosis. Bar = 100 μm. (E) ntATX-D2 vaccinated bird. The deep dermal collagen is separated by abundant fibrin and edema, with small numbers of scattered heterophils. Bar = 50 μm. (F) ntATX-D2 vaccinated bird. Severe cell lysis of erythrocytes within blood

vessels, with marked expansion of the adjacent deep dermis by abundant fibrin and edema. Bar = 50 μm.
(TIF)

**S4 Fig. Histopathological changes in the muscle of turkeys representing experimental groups.** (A) Negative control. No lesions were present within any of the negative control birds, with scores equivalent to 1 for all categories. Bar = 50 μm. (B) Positive control. Skeletal myofibers often exhibit monophasic sarcoplasmic fragmentation and hypereosinophilia with loss of cross-striations, affecting over 75% of the myofibers (lesion score = 5). Myofibers are often separated by fibrin and edema (lesion score = 4) admixed with scattered cellular lysis, affecting 30–75% of the tissue. An inflammatory infiltrate is absent. Bar = 50 μm. (C) Positive control. Foci of cell lysis, fibrin, and edema frequently contain abundant large bacterial rods. Lesion score was 4 for bacteria, with bacteria scattered extensively throughout 30–75% of the section. Bar = 10 μm. (D) ntATX-D1 vaccinated bird. Skeletal myofibers are replaced by a focally extensive caseous granuloma, characterized by an outer rim of abundant macrophages, multi-nucleated giant cells, and fewer heterophils and lymphocytes (granuloma lesion score = 4). The rim surrounds an inner core of caseous necrosis of skeletal muscle, with abundant cellular debris and brightly eosinophilic, necrotic myofibers (caseous necrosis lesion score = 4). Bar = 50 μm. (E) ntATX-D1 vaccinated bird. Erythrocytes within and surrounding vessels exhibit cell lysis multifocally in up to 25% of the section, consistent with a cell lysis lesion score of 3 (arrowhead). Myofibers exhibit similar degeneration and necrosis as in the positive controls. Fibrin and edema multifocally separate myofibers. Bar = 50 μm. (F) ntATX-D2 vaccinated bird. Similar cell lysis, fibrin, edema, and myopathy are visualized as described in the positive control and D1 vaccinated bird. Bar = 50 μm.
(TIF)

**S5 Fig. Histopathological changes in the spleen of turkeys representing experimental groups.** (A) Negative control. No lesions are identified with the negative control group. The arrowhead represents a bursa dependent lymphoid nodule. (B) Positive control. A subset of birds had unapparent bursal dependent nodules, depletion of lymphocytes around sheathed venules, and reticular hyperplasia. Scores in this animal for lymphoid depletion and reticular hyperplasia (arrow) were a 4 with 30–75% of the tissue involved by multifocal lesions. (C) ntATX-D1 vaccinated bird. Occasional birds had atrophy of bursal dependent lymphoid nodules (arrowhead) and of lymphocytes around sheathed venules (lymphoid depletion lesion score = 3). Reticular hyperplasia was equivalent of a lesion score 4, affecting 30–75% of the tissue. (D) ntATX-D2 vaccinated bird. Occasional birds had lymphoid depletion (arrowhead) and reticular hyperplasia (arrow). Bars = 50 μm.
(TIF)

## Acknowledgments

We thank Butterball LLC., North Carolina for providing the turkeys for use in this study. We acknowledge the Laboratory Animal Resources facility for animal housing, and the Flow Cytometry Core facility support for immunophenotyping at the College of Veterinary Medicine, North Carolina State University. We also thank Dr. Eliza Ripplinger, Ms. Mitsu Suyemoto and Mr. Varches Gotaparthy, for their assistance related to the animal work of this study.

## Author Contributions

**Conceptualization:** Anil J. Thachil, Rocio Crespo, Raveendra R. Kulkarni.

**Data curation:** Feba Ann John, Abigail Armwood, Raveendra R. Kulkarni.

**Formal analysis:** Feba Ann John, Valeria Criollo, Carissa Gaghan, Abigail Armwood, Raveendra R. Kulkarni.

**Funding acquisition:** Raveendra R. Kulkarni.

**Investigation:** Feba Ann John, Valeria Criollo, Carissa Gaghan, Abigail Armwood, Jennifer Holmes, Anil J. Thachil, Rocio Crespo, Raveendra R. Kulkarni.

**Methodology:** Feba Ann John, Valeria Criollo, Carissa Gaghan, Abigail Armwood, Jennifer Holmes, Anil J. Thachil, Rocio Crespo, Raveendra R. Kulkarni.

**Project administration:** Raveendra R. Kulkarni.

**Software:** Feba Ann John.

**Supervision:** Raveendra R. Kulkarni.

**Writing – original draft:** Feba Ann John, Valeria Criollo, Raveendra R. Kulkarni.

**Writing – review & editing:** Carissa Gaghan, Abigail Armwood, Jennifer Holmes, Anil J. Thachil, Rocio Crespo, Raveendra R. Kulkarni.

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
