## [Decision Letter · Decision Letter 0]

4 Mar 2024

PONE-D-23-44060Immunization of turkeys with Clostridium septicum alpha toxin-based recombinant subunit proteins can confer protection against experimental Clostridial dermatitis.PLOS ONE

Dear Dr. Kulkarni, Thank you for submitting your manuscript to PLOS ONE. After careful consideration, we feel that it has merit but does not fully meet PLOS ONE’s publication criteria as it currently stands. Therefore, we invite you to submit a revised version of the manuscript that addresses the points raised during the review process.

We look forward to receiving your revised manuscript.

Kind regards,

Yung-Fu Chang

Academic Editor

PLOS ONE

Journal Requirements:

4. We note that you have a patent relating to material pertinent to this article. Please provide an amended statement of Competing Interests to declare this patent (with details including name and number), along with any other relevant declarations relating to employment, consultancy, patents, products in development or modified products etc. Please confirm that this does not alter your adherence to all PLOS ONE policies on sharing data and materials, as detailed online in our guide for authors http://journals.plos.org/plosone/s/competing-interests by including the following statement: ""This does not alter our adherence to  PLOS ONE policies on sharing data and materials.” If there are restrictions on sharing of data and/or materials, please state these. Please note that we cannot proceed with consideration of your article until this information has been declared.

**Additional Editor Comments:**

Your manuscript has been reviewed by an expert and his comments are enclosed for your reference.  Please follow the comments and make all necessary revision. Please also answer all questions one by one.

Reviewers' comments:

Reviewer's Responses to Questions

**Comments to the Author**

1. Is the manuscript technically sound, and do the data support the conclusions?

Reviewer #1: Yes

2. Has the statistical analysis been performed appropriately and rigorously? 

Reviewer #1: Yes

3. Have the authors made all data underlying the findings in their manuscript fully available?

Reviewer #1: Yes

4. Is the manuscript presented in an intelligible fashion and written in standard English?

Reviewer #1: Yes

5. Review Comments to the Author

Reviewer #1: Overall, the manuscript is well written and the data is quite interesting. However, there seems to be lack of clarity over the induction of the different arms of the immune system by subunit vaccines. The way the authors approached the explanation of the results and conclusion, it’s not clear which (innate, cell or humoral) immunity has been induced. They can argue that all of them were induced, but testing for cytokines in vaccinated birds at termination (of experiment??) doesn’t sound rational. I would suggest that the authors clearly distinguish which arms of the immune system has been induced.

L56 – 57: Are you referring to a commercial availability here? Please be specific because research has been ongoing for a while to develop vaccine with some vaccines showing promise. Also, please specify here if you are only referring to vaccines for poultry.

L158: Please provide more information about the strain here. You have cited “5” but more information about how “hot” the strain is would be required. What is the expected mortality/morbidity when you challenge the birds with this strain. Please add this information

L192: At which time point were the tissues collected? I can only assume that you did qPCR on tissues collected at termination (but what does termination mean?). Is it when the birds died, or you had a specific age of birds? Please make this clear.

L242: For the ELISA section, what was the starting concertation of serum? Was it whole serum or you diluted it to a certain point first? It may be shown in your plots, please specify here. Also, I would ask the authors to consider using the following paper to do a statistical analysis of their ELISA data. You can also do End-Point titre method.

L257: Add the statistical analysis method you used for ELISA. Also, why didn’t you analyze your mortality? Can you do a survival analysis using Kaplan-Meier curves or hazard ratios? This is very important for the mortality data, which seems to be quite significant for your study.

L270 – 282: This reads like a Materials and Methods. You have described these already. So, focus on the outcomes.

L283 – L295: Same as Above

L297 – 302: Same as above. Why do you need to repeat this here?

L355: Here you are being selective on which treatment comparisons you’re reporting. For instance, for IL-1b, you compare the two controls, but for IL-6, you only compare NCx and D2-immunized birds. If you see IL-6, PCx and the immunized groups don’t seem to show significant differences. When you look at the figure for NCx and PCx, the bars are quite different, and you statistic should show significance. This is relevant because your immunization may not be the cause for higher IL-6.

Suggestion: NCx vs PCx comparison only shows you if you challenge worked. The comparing immunization to PCx should be reported. Here letter based identification for statistical significance may be useful. Please to make the best out of your data.

L460: Figure 8F not shown on figure

L480 – 484: Please revise after you consider the statistical analysis

L535 – 538: Good point. But you didn’t show this in the figure, not did you describe it in your “results” section

L550 – 553: Compared to which treatment? Be specific

L559 – 562: This conclusion may be a bit of a stretch. I can argue that because you’re analyzing innate immune response genes at “termination” (if my assumption is correct, this would be post-vaccination), then your lower innate genes could be manily because the birds are mounting strong antibody mendiated responses, hence not require innate mediated response. Given the fact that you’re using a subunit vaccine, this is highly likely. Please revise

L594 – 600: Mortality is one of your strongest data. Why did you omit it from your overall conclusion?

6. PLOS authors have the option to publish the peer review history of their article (what does this mean?). If published, this will include your full peer review and any attached files.

Reviewer #1: No

---

## [Author Response · Author response to Decision Letter 0]

22 Mar 2024

Response to reviewer document has been uploaded as a separate file as well as included in the rebuttal letter to the Editor.

---

## [Decision Letter · Decision Letter 1]

9 Apr 2024

Immunization of turkeys with Clostridium septicum alpha toxin-based recombinant subunit proteins can confer protection against experimental Clostridial dermatitis.

PONE-D-23-44060R1

Dear Dr. Kulkarni,

We’re pleased to inform you that your manuscript has been judged scientifically suitable for publication and will be formally accepted for publication once it meets all outstanding technical requirements.

Kind regards,

Yung-Fu Chang

Academic Editor

PLOS ONE

Additional Editor Comments (optional):

Reviewers' comments:

Reviewer's Responses to Questions

**Comments to the Author**

1. If the authors have adequately addressed your comments raised in a previous round of review and you feel that this manuscript is now acceptable for publication, you may indicate that here to bypass the “Comments to the Author” section, enter your conflict of interest statement in the “Confidential to Editor” section, and submit your "Accept" recommendation.

Reviewer #1: All comments have been addressed

2. Is the manuscript technically sound, and do the data support the conclusions?

Reviewer #1: Yes

3. Has the statistical analysis been performed appropriately and rigorously? 

Reviewer #1: Yes

4. Have the authors made all data underlying the findings in their manuscript fully available?

Reviewer #1: Yes

5. Is the manuscript presented in an intelligible fashion and written in standard English?

Reviewer #1: Yes

6. Review Comments to the Author

Reviewer #1: (No Response)

7. PLOS authors have the option to publish the peer review history of their article (what does this mean?). If published, this will include your full peer review and any attached files.

Reviewer #1: No
